# Optimization of Imputation Strategies for High-Resolution Gas Chromatography–Mass Spectrometry (HR GC–MS) Metabolomics Data

**DOI:** 10.3390/metabo12050429

**Published:** 2022-05-11

**Authors:** Isaac Ampong, Kip D. Zimmerman, Peter W. Nathanielsz, Laura A. Cox, Michael Olivier

**Affiliations:** 1Center for Precision Medicine, Department of Internal Medicine, Section on Molecular Medicine, Wake Forest University, Winston-Salem, NC 27157, USA; iampong@wakehealth.edu (I.A.); kdzimmer@wakehealth.edu (K.D.Z.); laurcox@wakehealth.edu (L.A.C.); 2Center for the Study of Fetal Programming, University of Wyoming, Laramie, WY 82071, USA; peter.nathanielsz@uwyo.edu; 3Southwest National Primate Research Center, San Antonio, TX 78227, USA

**Keywords:** metabolomics, HR GC–MS, imputation missing values

## Abstract

Gas chromatography–coupled mass spectrometry (GC–MS) has been used in biomedical research to analyze volatile, non-polar, and polar metabolites in a wide array of sample types. Despite advances in technology, missing values are still common in metabolomics datasets and must be properly handled. We evaluated the performance of ten commonly used missing value imputation methods with metabolites analyzed on an HR GC–MS instrument. By introducing missing values into the complete (i.e., data without any missing values) National Institute of Standards and Technology (NIST) plasma dataset, we demonstrate that random forest (RF), glmnet ridge regression (GRR), and Bayesian principal component analysis (BPCA) shared the lowest root mean squared error (RMSE) in technical replicate data. Further examination of these three methods in data from baboon plasma and liver samples demonstrated they all maintained high accuracy. Overall, our analysis suggests that any of the three imputation methods can be applied effectively to untargeted metabolomics datasets with high accuracy. However, it is important to note that imputation will alter the correlation structure of the dataset and bias downstream regression coefficients and *p*-values.

## 1. Introduction

Metabolomics describes the systematic identification and quantification of a wide range of small molecules (<1500 Da) in biological samples (cells, tissues, biological fluids, etc.). Although the metabolomics field is relatively new compared to other omics fields such as genomics and proteomics, it has seen steady growth in recent years. This is due, at least in part, to the rapid development and implementation of new technology platforms in mass spectrometry (MS) [1], in addition to nuclear magnetic resonance (NMR) spectroscopy [2]. Generally, metabolomics studies utilize targeted and untargeted analyses as the two main approaches. Untargeted studies tend to focus on global identification and quantification of as many metabolites as possible, while targeted approaches aim to characterize and quantify a select set of known metabolites with higher accuracy and sensitivity [1].

One of the technologies used in metabolomics is gas chromatography–coupled mass spectrometry (GC–MS), which is capable of analyzing volatile, non-polar, and polar metabolites, often with the intent to uncover novel biomarkers that can advance our understanding of biological processes and molecular mechanisms in health and disease [3,4]. Among the many areas where MS-based metabolomics have been applied include precision medicine [5], forensic medicine [6], environmental [7], and food and nutritional sciences [8]. With the large number of metabolites captured and the wide dynamic range of intracellular metabolite concentrations, missing values are not uncommon in metabolomics datasets. However, in order to perform statistical analysis or combine such data with other omics datasets without simply removing all of the missing data points (which can lead to bias or reduce the number of metabolites to a small handful), the issue of missing values needs to be addressed [9]. Missing values in metabolomics can occur for both biological and technical reasons [10,11,12]. The three main types of missing values are (a) missing not at random (MNAR), (b) missing at random (MAR), and (c) missing completely at random (MCAR). Random missingness related to data acquisition processes such as incomplete derivatization or incomplete ionization of analytes is defined as MCAR. Unlike MCAR, MAR values are dependent on other observed variables [13]. An example of this is the interference of high concentrations of one metabolite causing other metabolites of lower concentration to go undetected. The missing values of metabolites with low concentrations below the analytical platform’s limit of detection (LoD) are categorized as MNAR [1,14]. It is challenging to determine the underlying mechanism(s) of missingness that exist in a dataset as there is not yet a rigorous, straightforward method to diagnose and distinguish these mechanisms [15], and there is no accepted approach to correct the analysis of data once the causal mechanism has been defined. Advancements in analytical platforms and improvements in bioinformatics tools have provided partial solutions to reducing missing values in metabolomics data [16]. Several imputation algorithms have been established that can handle missing values in metabolomics data, but there is no consensus on which imputation method performs best on metabolomics data. In the past, imputation methods including random forest (RF), k-nearest neighbors (kNN), local least squares (LLS), half minimum, average, singular value decomposition (SVD), generalized ridge regression (GRR), zero, quantile regression (QR), and Bayesian principal component analysis (BPCA) have been applied to impute missing metabolomics data [13,15]. RF generally handles both parametric and non-parametric data sets of complex linear and non-linear problems and normally does not need preprocessing of data. In contrast, BPCA performs better if the data are transformed first before imputation but is dependent on dimension reduction. Lastly, the KNN imputation technique can be used for both discrete and continuous data but has been reported to have low precision when imputing variables and introduces false associations where they do not exist [17]. While these advanced methods, including KNN, SVD, BPCA, and RF, have been reported to be good for imputing MCAR/MAR, they are generally considered to be unsuitable for imputing data with MNAR [3]. Determined value substitution methods such as Zero, Mean, and HM are fast and simple methods for handling a limited set of missing values in a dataset, but they underestimate variance, ignore the relationship with other variables, and bias summary statistics [13]. Regression methods such as GRR and QR can preserve their correlation with other variables and are generally considered to be suitable for imputing MNAR, but the variability of missing values is underestimated [13,18]. Finally, LLS methods are known to estimate missing values accurately if the data matrix contains rich local structure but perform poorly when the missing values have a strong correlation with the global rather than local structure [19].

So far, only a few studies have compared the performance and assessed the accuracy of these various imputation methods on metabolomics data, and these analyses have all been limited to real biological datasets. Hence, our major motivation for this study was to systematically evaluate the performance of imputation methods with a dataset obtained from NIST plasma technical replicates, which were acquired under the same MS conditions over time. In theory, these datasets should have a similar data structure (e.g., they are similar in terms of number and patterns of missing data and share the mechanism underlying missingness of the data) before further validation in real biological samples. Given the additional sample preparation steps needed to derivatize samples in preparation for GC–MS, the data often show greater variability when compared to liquid chromatography–mass spectrometry (LC-MS) data. Therefore, it is important to assess the performance of imputation algorithms specifically for GC–MS data and use a dataset of replicate samples that, in theory, should have minimal variation. Missing values may either mask, or be directly related to, a biological response, and it is important to evaluate the accuracy of missing data imputation and the impact of the imputed values on downstream statistical correlation and association analyses.

In this study, we evaluated the performance of multiple imputation strategies for untargeted high resolution (HR) GC–MS-based metabolomics datasets. This was initially completed by using technical replicates of the National Institute of Standards and Technology (NIST) Standard Reference Material (SRM) 1950 human plasma metabolomics data generated by an HR GC–MS. With the complete NIST plasma dataset, we introduced missingness with four missing mechanisms (MCAR, MAR, MNAR, and a combination of all three: MCAR–MAR–MNAR) at various proportions of missingness (2–70%) and subsequently evaluated the accuracy of 10 imputation methods. For the most accurate imputation methods, we then re-evaluated the imputation accuracy and bias in correlation and downstream statistical analyses in two different datasets (GC–MS metabolomics data from plasma and liver) collected from non-human primates (NHP) Figure 1.

## 2. Results

To systematically assess the accuracy of 10 imputation methods commonly applied in metabolomics studies, replicates of a commercially available NIST plasma sample, baboon plasma, and liver samples were processed and acquired on a Q-Exactive HRGC Orbitrap-MS in electron ionization (EI) mode. Following data analysis and metabolite identification, raw data were separated into either full data (with missing datapoints for individual metabolites/samples) or complete data without any missing data. Missing values were incorporated into complete or, in some cases, full datasets using various missingness types at different missing proportions and with various coefficients of variation (CV) prior to imputation method evaluations (Figure 1).

### 2.1. Missing Values Simulation and Imputation Evaluation Using HR GC–MS Metabolomics Data for Replicates of NIST Plasma

In order to systematically evaluate and assess the performance of various imputation methods using metabolomics data acquired with an HR Orbitrap GC–MS, ten common imputation methods (RF, GRR, BPCA, Mean, LLS, SVD, QR, kNN, HM, and Zero) were initially evaluated for accuracy in the complete NIST plasma dataset with four different missing mechanisms (i.e., MCAR, MAR, MNAR, and MCAR–MAR–MNAR). Evaluation of imputation accuracy with RMSE revealed four methods (RF, GRR, BPCA, and Mean) with comparably low RMSEs (Figure 2). The RMSEs of these methods showed similar patterns across all four missing value types (Appendix A). Further, to evaluate imputation methods on the full NIST plasma dataset (i.e., the entire dataset, including the real missing values), we imputed with Mean, BPCA, GRR, and RF and assessed imputation performance on these technical replicate samples based on correlation and Cronbach’s alpha. Here, we would expect the mean sample-to-sample correlation to be as close to 1 as possible. The Cronbach’s alpha is a measure of internal consistency that is highly related to the overall correlation structure. After imputation, RF, GRR, and BPCA all produced the highest mean sample-to-sample correlations and Cronbach’s alphas. The mean sample-to-sample correlations are listed along with the respective Cronbach’s alpha in parenthesis for each method: RF 0.70 (0.986), GRR 0.70 (0.986), BPCA 0.70 (0.988), Mean 0.67 (0.985).

### 2.2. Evaluation of RF, GRR, and BPCA Imputation Methods on NHP Plasma

The three most accurate imputation methods (RF, GRR, and BPCA) were subsequently used for imputation of metabolomics data from baboon plasma samples. Using a mixture of all three missingness types (MCAR–MAR–MNAR), we evaluated imputation accuracy based on RMSE at various ranges of missingness and different coefficients of variance (CV) in the complete metabolomics dataset obtained for baboon plasma samples. We found significant increases in accuracy when imputing with GRR over the other two methods for CVs above 0.2 (Figure 3). To assess the correlation between RMSE and CV, we also performed a simple linear regression analysis with the imputation method adjusted as the covariant and revealed a significant association between CV and RMSE (estimate = 1.48, *p* = 3.9 × 10^−13^). Similarly, metabolite missingness demonstrated a significant positive association with RMSE (estimate = 0.44, 4.2 × 10^−15^).

In most scenarios, BPCA was generally the least accurate method. The evaluation of Cronbach’s alpha to assess the differences between the imputed and real baboon plasma metabolomics data revealed that BPCA increased the correlations among the imputed results. Meanwhile, the GRR and random forest imputations decreased the correlations below truth. No significant differences were noted between random forest and GRR (Figure 4). To test for biases in downstream statistical analysis, we estimated the differences between the regression coefficients of the imputed metabolites and the regression coefficients of the complete, original metabolite data. Regression analysis for this study was computed as a function of age, where age was the predictor and the metabolite abundance was the outcome. Pairwise Wilcoxon comparisons of the differences between the true and imputed regression estimates for each method revealed statistically significant differences when applying the random forest imputation method (Figure 4).

When using a one-sample *t*-test to test for departure from zero in the differences between the true and imputed regression estimates, all methods showed small but significant downward biases (Figure 5). We observed similar patterns for each of the methods across all different types and proportions of missingness (data not shown). These results indicate that, on average, imputation is decreasing the magnitude of effect sizes. This is to be expected because when a relationship between a variable exists and then missingness is introduced, the imputation will generally fail to recapture the full strength of that relationship. This will be particularly true for imputation in such a high-dimensional sample space where many variables influence the imputation of a particular value. Using a one-sample *t*-test to test for departure from zero in the differences between the true and imputed *p*-values, all of the methods showed a significant downward bias (FDR *p* = 0.02, 0.02, 0.004 for GRR, BPCA, RF, respectively). GRR demonstrated the smallest shift in these *p*-value differences, but all of the methods had mean differences that were slightly below zero. Again, this indicates that, on average, the *p*-values after imputation are larger (less significant) than they were prior to introducing missing values. It is worth noting, however, that while these general trends occur, there are still frequent occasions where the *p*-values were much smaller after imputation for individual metabolites.

### 2.3. Evaluation of RF, GRR, and BPCA Imputation Methods Using Metabolomics Data from Baboon Liver Biopsy Samples

Having evaluated these three imputation methods in baboon plasma metabolomics data, we then performed a similar analysis on metabolites extracted from liver tissue samples to assess if the same patterns hold when using a different dataset such as tissue-derived metabolites. The imputation accuracy evaluated at various missingness thresholds and CVs in the complete baboon liver samples revealed that as both the CVs and the proportions of metabolite missingness increase, GRR and RF appear to be the most accurate methods (Figure 6). A simple linear regression analysis between RMSE and CV revealed a significant positive correlation between CV and RMSE that holds across all three methods (estimate = 1.42, *p* = 9 × 10^−8^). Similarly, metabolite missingness demonstrated a significant positive association with RMSE (estimate = 0.42, 2.7 × 10^−12^). 

The differences in accuracy between the three methods are marginal, but significant in some cases. Consistent with the baboon plasma metabolomics dataset, the differences between the Cronbach’s alpha computed on the complete dataset, and the imputed liver dataset revealed that BPCA increased the correlation above truth, while GRR and random forest decreased the correlation below truth. No significant differences were noted between random forest and GRR (Figure 7). Pairwise Wilcoxon comparisons revealed no statistically significant differences between methods in their differences between the true and imputed regression estimates. A one-sample *t*-test to test for departure from zero in the differences between the true and imputed regression estimates showed a small but significant downward bias in the liver dataset, similar to that reported for the plasma dataset (Figure 7). This was true for each of the methods across all of the different types and proportions of missingness (data not shown). In terms of computing the true and imputed *p*-value differences using a one-sample *t*-test to test for departure from zero, we reported that only the BPCA method showed a significant downward bias (FDR *p* = 0.11, 7.3 × 10^−9^, 0.96 for GRR, BPCA, RF, respectively). Random forest demonstrated the smallest shift in these *p*-value differences, but all of the methods had mean differences that were slightly below zero (Figure 7).

### 2.4. In-Depth Evaluation of RF Imputation Accuracy at Wide Range of Missingness Using the Entire Baboon Liver HR GC–MS Metabolomics Dataset

We further examined RF imputation on the full dataset with a wider range of missingness to assess the limits of meaningful imputation of missing data. The goal of these analyses was to observe the behavior of RF imputation on data where the global data structure is as close to reality as possible. To minimize altering the global data structure, we iteratively inserted only a single missing value into the existing data for a particular metabolite where there was previously a real value. The full data were then imputed, but only the difference between the original value and the imputed value at that one data point was recorded along with the information about the metabolite in which the missing data point was introduced (e.g., CV, proportion of already missing values, etc.). These analyses were intended to determine the proportion of missing values in metabolomics data where imputation begins to introduce strong biases or becomes unreasonably inaccurate. The majority of the percentage bias (PB) values were below 5%. However, there was an increase in percentage bias (i.e., decreased accuracy) in metabolites with more than 45% missing data (Figure 8). The evaluation of Cronbach’s alpha showed that at a low proportion of overall missingness, there were small but significant increases in the mean correlations between metabolites when imputing with random forest. However, as missingness increased, this pattern diminished and then actually reversed where the correlations between metabolites decreased when imputing with random forest. We also observed a mean downward bias in the regression coefficients that grew larger as the amount of missingness increased. In addition, we observed that with more missingness, the standard deviation of the differences between the regression coefficients from imputed data and the real values also increased. Finally, there was a slight mean upward bias in the *p*-values, which grew larger as the amount of overall missingness increased (Figure 8).

## 3. Discussion

The metabolomics field continues to advance, primarily due to technological platform improvements and advances in data analysis, but the issue of missing values in metabolomics datasets still remains a challenge [1]. Simply removing missing values or imputing missing values with sub-optimal methods can have dire consequences on downstream statistical analysis [20,21]. In the past, several studies have evaluated the performance of imputation methods, particularly using LCMS data [15]. Generally, evaluations of different imputation methods for missing data in metabolomics datasets have been based on simulated datasets or limited biological or clinical study data [1,22,23]. We performed a systematic evaluation of imputation accuracy and bias-based analysis first on technical replicate data from a standard sample and then, further, on the performance of the top three imputation methods commonly cited in the literature, on baboon plasma and liver tissue metabolomics data. We utilized metabolomics datasets acquired by an HR Q-Exactive GC–Orbitrap MS, which, to the best of our knowledge, is the first time such a thorough imputation analysis using this newest metabolomics platform has been performed.

Ten imputation methods were initially evaluated for accuracy in the complete NIST plasma dataset, where we introduced four different types of missing values (MCAR, MAR, MNAR, and a combination of all three: MCAR–MAR–MNAR). Because the NIST plasma sample dataset included technical replicates, the correlation across samples should be very high. Therefore, we made the assumption that imputation methods that increased mean sample-to-sample correlation would be the most optimal. In the NIST plasma GC–MS dataset, we observed that GRR, RF, and BPCA were all highly accurate and all improved the sample-to-sample correlation between technical replicates. These patterns held for all types of missing data regardless of the assumed mechanism that causes them. However, it is important to note that in our current study LoD, related methods such as HM and zero performed poorly on MNAR simulated data, which contradicts findings from previous studies [11,12]. These differences in findings may be, in part, because of the differences in the nature of data used for simulation. We used technical replicates to assess imputation algorithm performance, whereas prior studies exclusively used data from biological replicates [11,12]. We further evaluated these methods for accuracy and bias in the plasma and liver samples of healthy adult baboons. We pursued these analyses in this order to specifically examine how imputation methods perform as one moves from an “ideal” metabolomics dataset to more realistic plasma samples from NHP and then even into potentially less consistent metabolomics samples extracted from tissue.

The analysis of the adult baboon plasma and liver tissue metabolomics datasets emphasized that there does not appear to be a single, superior imputation method. However, among the three most highly accurate methods observed here, GRR and RF showed improved performance relative to BPCA for some of the assessments. It is worth noting, though, that the differences between these methods are very minor, and these differences may change depending on the dataset that is being imputed, as we observe to a small degree here. In both plasma and liver metabolomics datasets, there was a significant downward or upward bias, as demonstrated by Cronbach’s alpha, percent bias, *p*-value, and regression estimates in all three methods. This means that no matter what imputation method (unless one is applying multiple imputation) is applied in metabolomics data, it will alter the correlation structure, regression coefficients, and *p*-values. As we have demonstrated in our current study, more often than not, imputation will shrink regression coefficients, and it is important to note that, in general, such biases grow even stronger with increased proportions of metabolite missingness. For this reason, we suggest that imputation should always be applied with caution when making statistical inferences in downstream analyses and interpretations.

While none of the top three methods in this study show better overall performance than any of the others, we would like to note that random forest imputation provides a robust approach to impute missing data from HR GC–MS untargeted metabolomics datasets. Random forest imputation has also been highlighted in a previous LC–MS study, where it was suggested as the best method for imputing missing data in metabolomics [15]. Primarily, the RF method benefits from being a non-parametric technique that does not make distributional assumptions about the data [24]. This is important in metabolomics data, where the distributions for individual metabolites are not easily categorized and are likely variable. We note that beyond 40% of metabolite missingness, we do not recommend random forest imputation (or any imputation), as imputation will lose accuracy and introduce stronger and more significant biases into data, thereby affecting downstream statistical analyses and interpretation of results. When regressing RMSE on either CV or metabolite missingness, we identified strong positive associations. As expected, these results imply that imputation accuracy deteriorates as both metabolite missingness and CVs increase. Although it is already common practice to filter out such metabolites, we re-emphasize that computing analysis with metabolites that have higher CVs and proportions of missingness should be done with caution. Metabolites with large proportions of missingness should not be considered for statistical analysis, except in the case where one is interested in testing differential rates of missingness between groups (i.e., testing if a particular metabolite is missing more frequently from one group because of a potential biological phenomenon).

This study systematically investigates multiple possible underlying mechanisms leading to missing data in metabolomics experiments and uses both technical replicates of a reference sample and sets of real biological samples to determine the best imputation methods for GC–MS-based metabolomics studies. Despite this comprehensive approach, there are still limitations, some of which will need to be addressed in future imputation studies. The current study was limited to only untargeted HR GC–MS metabolomics analyses. It is important for future studies to extend this to targeted HR GC–MS and other metabolomics studies, although it is important to note that the RF imputation approach has also been recommended for the imputation of LC–MS data [15]. Moreover, our study focused on comparing the accuracy of imputation methods using imputed data generated from technical replicates and real biological datasets, and future studies will need to extend these analyses to better understand the impact of imputation on downstream data analysis and biological interpretation of data. Improper handling of missing values is known to bias subsequent downstream statistical analysis, including multivariate analysis [12,25,26]. We would also like to emphasize that all methods examined in this study were single imputation-based methods. We recognize the many benefits of multiple imputations to provide unbiased results in downstream statistical analyses [27,28,29]. However, applying multiple imputations in a high-dimensional setting is complex and exceeds the scope of this manuscript. It is also likely to be a computationally challenging approach to implement for most metabolomics researchers; therefore, we restricted our analysis to methods widely available to scientists in the field.

Unfortunately, for most metabolomics datasets, it is impossible to know what type of missingness and how much of each type of missingness exists in the data. For example, if the vast majority of the missing values in the data are only MNAR because the metabolites that are missing are missing simply due to low abundance, then our results here will be biased. This is particularly true because we started with the complete data (where there were no missing values) and then added missingness from there. Even though we introduced missingness in a way that would be consistent with MNAR (see Methods), we were imputing back to and comparing against highly abundant values that were really there in the first place, not metabolites that were actually low abundance metabolites. It is worth noting, though, that the NIST plasma dataset contained missing values in metabolites that are actually highly abundant in some technical replicates and missing in other samples that should be identical. This indicates that missingness is not due only to low abundance, but can occur for other reasons, too. Lastly, we note that imputation is always a guess (no matter how informed it may be), and it must be applied carefully. Any discoveries made by way of imputation should always be validated.

## 4. Materials and Methods

### 4.1. Chemicals and Reagents

All solvents for GC–HR Orbitrap MS analysis (HPLC-grade acetonitrile, isopropanol, pyridine, methanol) were purchased from Sigma-Aldrich (St. Louis, MO, USA). Consumables such as syringes, liners, septa, and filament were purchased from Thermo Scientific (Madison, WI, USA). GC amber vials, caps, tubes with inserts were purchased from Microsolv Technology Corporation (Greater Wilmington, NC, USA).

For this imputation study, we used three independent metabolomics datasets that were generated using an HR-GC-Orbitrap-MS: the NIST SRM 1950 human plasma purchased from Sigma-Aldrich (St. Louis, MO, USA) and data obtained from NHP (baboon) plasma and liver samples while animals were fed a normal chow diet.

### 4.2. Sample Processing

Metabolite extraction from NIST plasma and baboon plasma and liver was adapted from a previously described protocol [30]. In brief, aliquots (15 μL) of plasma or liver samples were subjected to sequential solvent extraction, once each with 1 mL of acetonitrile: isopropanol: water (3:3:2) and 500 μL of acetonitrile: water (1:1) mixtures at 4 °C [31]. An internal standard, adonitol (2 μL from 10 mg/mL stock), was added to each aliquot prior to the extraction. The extracts were dried under vacuum at 4 °C prior to chemical derivatization (silylation reactions). Blank tubes without samples were treated similarly to sample tubes and added to account for background noise and other sources of contamination. Samples and blanks were sequentially derivatized with methoxyamine hydrochloride (MeOX) and 1% TMCS in N-methyl-N-trimethylsilyl-trifluoroacetamide (MSTFA) or 1% TMCS containing N-(t-butyldimethylsilyl)-N-methyltrifluoroacetamide (MTBSTFA), as described elsewhere [32]. Briefly, the steps involved addition of 20 μL of MeOX (20 mg mL^−1^) in pyridine incubated at 55 °C for 60 min, followed by trimethylsilylation at 60 °C for 60 min after adding 80 μL MTBSTFA.

### 4.3. GC-HR Orbitrap MS Data Acquisition and Preprocessing

Metabolites were analyzed by high-resolution/accurate (HRAM) Orbitrap mass spectrometry (Q-Orbitrap MS, Thermo Fisher) coupled to gas chromatography (GC). In all cases, 1 µL of derivatized sample was injected into the TRACE 1310 Gas chromatography (Thermo Scientific, Austin, TX, USA) in a splitless (SSL) mode at 220 °C. The carrier gas was helium set at a flow rate of 1 mL/min for separation on a Thermo Scientific Trace GOLD TG-5SIL-MS 30 m length × 0.25 mm i.d. × 0.25 μm film thickness column with an initial oven temperature of 50 °C for 0.5 min, followed by an initial gradient of 20 °C/min ramp rate. The final temperature was 300 °C and was held for 10 min. Eluting peaks were transferred through an auxiliary transfer line temperature of 230 °C into a Q Exactive-GC mass spectrometer (Thermo Scientific, Bremen, Germany). The total run time was 25 min. Data were acquired in electron ionization (EI) mode at 70 eV energy, emission current of 50 μA with an ion source temperature of 250 °C. A filament delay of 5.7 min was selected to prevent excess reagents from being ionized. High-resolution EI spectra were acquired using 60,000 resolution (fwhm at *m/z* 200) with a mass range of *m/z* 50–650. The transfer line was set to 230 °C, and the ion source was set to 250 °C. Data acquisition and instrument control were carried out using Xcalibur 4.3 and TraceFinder 4.1 softwares (Thermo Scientific). The capillary voltage was 3500 V with a scan rate of 1 scan/s. Finally, raw data (.raw files) obtained from data acquired by GCMS were converted to .mzML formats using ProteoWizard’s msConvert tool prior to data preprocessing using open source software, MS-DIAL 4.6 (Riken, Japan, and Fiehn Lab, UC Davis, Davis, CA, USA). The MS-DIAL 4.6 was used for raw peak extraction, and the data baseline filtering and calibration of the baseline, peak alignment, deconvolution analysis, peak annotation, and integration of the peak height essentially followed as described [33]. Key parameters used include a peak width of 20 scan and minimum peak height of 10,000 amplitudes was applied for peak detection, and sigma window value of 0.5, EI spectra cutoff of 50,000 amplitudes was implemented for deconvolution. For the annotation setting, the retention time tolerance was 0.5 min, the *m/z* tolerance was 0.5 Da, the EI similarity cutoff was 60%, and the annotation score cutoff was 60%. In the alignment parameters setting process, the retention time tolerance was 0.5 min, and the retention time factor was 0.5. The detailed data preprocessing parameters were the same for all metabolomics datasets and are found in Appendix A. Spectral library matching for metabolite identification was performed using an in-house and public library consisting of pool EI spectra from MassBank, GNPS, RIKEN, MoNA. All data were normalized by QC-based LOESS normalization followed by log_10_ transformation.

The NIST plasma metabolomics dataset consisted of 150 replicate samples which were acquired in 12 different batches using an untargeted EI–GC–MS approach. The 12 batched datasets were pooled, aligned, and processed using open source software MS-DIAL (v4.6) [34]. A complete dataset of identified and quantified metabolites containing only metabolites with no missing values was created from the full dataset. The complete dataset was used to evaluate the accuracy of each of the imputation methods. The full dataset with missing values in some metabolites was used to assess which imputation method improved sample-to-sample correlation between technical replicates the most.

The second dataset was generated from metabolic profiling of 45 baboon plasma samples collected from 35 females in the age range of 6–23 years and 10 males in the same age range. All 45 plasma samples were analyzed using an untargeted EI–GC–MS approach, as described above.

The third dataset consists of another EI–GC–MS analysis of metabolites extracted from 47 liver biopsy samples collected from the same adult healthy baboons as the plasma, which included 39 females and 8 males in the age range of 6–23 years. The metabolite extraction and data processing followed as previously described above.

### 4.4. Generation of Missing Values

For the initial evaluations of imputation accuracy and performance, only the complete NIST plasma data (i.e., metabolites with no missingness) were used. This dataset included complete information on 150 replicate samples with no missing data for 60 metabolites. Individual missing data were subsequently introduced to the complete data at varying rates and modeling four different mechanisms—missing completely at random (MCAR), missing at random (MAR), missing not at random (MNAR), and mixture of all three (MCAR–MAR–MNAR). Methods for simulating missingness have been previously described [11]. MCAR missing values were introduced simply by removing values at random from the complete dataset at different proportions of missingness. MNAR values were introduced to randomly selected metabolites using a chi-squared distribution to sample a percentage of missingness to introduce to that particular metabolite. From there, all of the lowest values within a randomly selected metabolite were made missing until the desired percentage of missingness was achieved. MAR values were introduced by jointly sorting two randomly selected metabolites based on the abundances of only one of the two metabolites. After sorting both metabolites, a randomly drawn percentage of values from the other metabolite not used to do the sorting were removed. Starting with the values where the abundances of the metabolite used for sorting were highest, we removed values until the percentage of drawn missingness was achieved. This was done to simulate a situation where high abundance in one metabolite leads to missingness in another metabolite. For the mixture of all three missingness mechanisms, each missing type was sequentially added to the data at equal proportions until the desired rate of missingness was achieved. The mixture of all three missingness mechanisms (MCAR–MAR–MNAR) was considered most heavily of all the simulations because all three types of missingness are likely present in most HR GC–MS data, with all influencing the imputation procedures.

### 4.5. Evaluation of Imputation Methods

Ten imputation methods were evaluated for accuracy in the complete NIST plasma dataset. The method names and respective run parameters for each of the ten imputation methods are provided in Appendix A. Percent bias (i.e., the percent difference between the imputed value and truth, i.e., the original value in the dataset) and RMSE were observed for each method, for each type of missingness, at different coefficients of variance, and for varying degrees of metabolite missingness or global missingness. We also examined the correlation structure after imputation in the full dataset of NIST plasma samples. Because the NIST plasma samples were technical replicates, it is expected that the correlation across each sample should be very high. Therefore, we made the assumption that imputation methods that increased the mean sample-to-sample correlation the most are better.

We selected three highly accurate methods (GRR, BPCA, and random forest) for further evaluation for accuracy and bias in the plasma and liver biopsy samples of healthy, aging baboons. Percent bias and RMSE were computed on each of the three datasets. Data were reduced to only metabolites with no missing data present (i.e., the complete data) before missing data were purposefully introduced. The data were then imputed with each of the methods of interest, and the differences between the imputed result and the true data were used to compute Percent bias and the RMSE as commonly computed when evaluating imputation methods.

For the comparison of regression coefficients, we first introduced missing values into the complete dataset, and then the data were merged with the available age information for each sample, and, subsequently, the missing values were imputed. A simple regression analysis was computed between age and each metabolite’s abundance, treating the imputed values as truth. The differences between the regression coefficients and *p*-values from models using the complete dataset and regression coefficients from the models using the imputed data were recorded for each iteration of missingness that was introduced into the complete dataset. This was computed for global missingness rates of 10% and 20% and each of the four missingness types. For each method, we computed pairwise Wilcoxon Rank Sum tests to compare each of the methods and a one-sample *t*-test on the differences between the imputed and true regression coefficients and *p*-values to test for departures from zero.

We also compared the Cronbach’s alpha (a measure of internal consistency) between the imputed results and the truth. The Cronbach’s alpha was computed with the complete dataset, and then missing values were introduced and imputation completed. Cronbach’s alpha was again computed on the imputed results and compared with the true Cronbach’s alpha. The difference between the two measures was recorded, and the process was repeated iteratively for global missingness rates of either 10% or 20% and each of the four missingness types. For each method, we computed pairwise Wilcoxon Rank Sum tests to compare each of the methods and a one-sample *t*-test on the differences between the imputed and true Cronbach’s alphas to test for departure from zero.

A more in-depth evaluation of accuracy and bias was carried out for the random forest method in the liver samples to determine the amount of missingness at which the method’s accuracy starts to deteriorate and where it begins to introduce biases, such as increased or decreased correlations. For this evaluation of accuracy, we attempted to maintain the data structure as close to reality as possible. We used the full dataset after removing metabolites with >75% missing data and randomly selecting only a single non-missing data point at a time, setting it to missing, and then imputing it along with all of the other already present missing values. This was done iteratively for 10,000 of the possible 15,620 non-missing values present in the data. For each iteration, the true value was recorded along with the imputed value as well as the percentage of missing data in the metabolite being imputed. These data were analyzed to determine if there was a clear drop-off in the accuracy of imputation as the proportion of missing values increases for a specific metabolite. In addition to these evaluations, we also computed Cronbach’s alpha and regression coefficients for the random forest method. These were computed in an identical fashion to how they were computed for all methods, just across wider ranges of overall missingness rates (10–70%).

## 5. Conclusions

Our study demonstrates that three imputation methods (RF, GRR, and BPCA) can be applied with high accuracy to untargeted HR GC–MS-derived metabolomics datasets. While the imputation of missing data was highly accurate, imputation altered the data correlation structure and biased both downstream regression coefficients and *p*-values. These biases grow stronger with increasing proportions of metabolite data missingness. Therefore, we suggest that imputation should only be applied when necessary and should always be applied with caution when making statistical inferences downstream.

## Figures and Tables

**Figure 1 metabolites-12-00429-f001:**
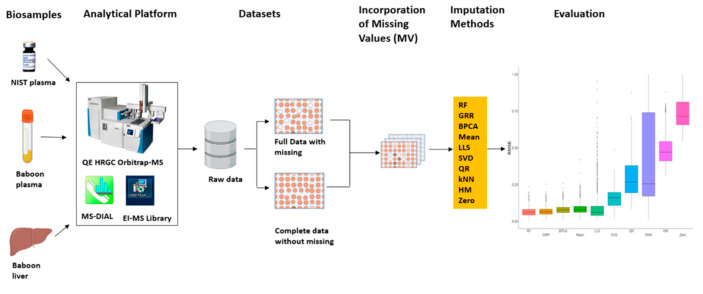
Metabolomics imputation study workflow. Diagram detailing metabolomics sample analysis, evaluation of imputation methods in technical replicate dataset (NIST plasma), and further validation in real baboon plasma and liver metabolomics datasets.

**Figure 2 metabolites-12-00429-f002:**
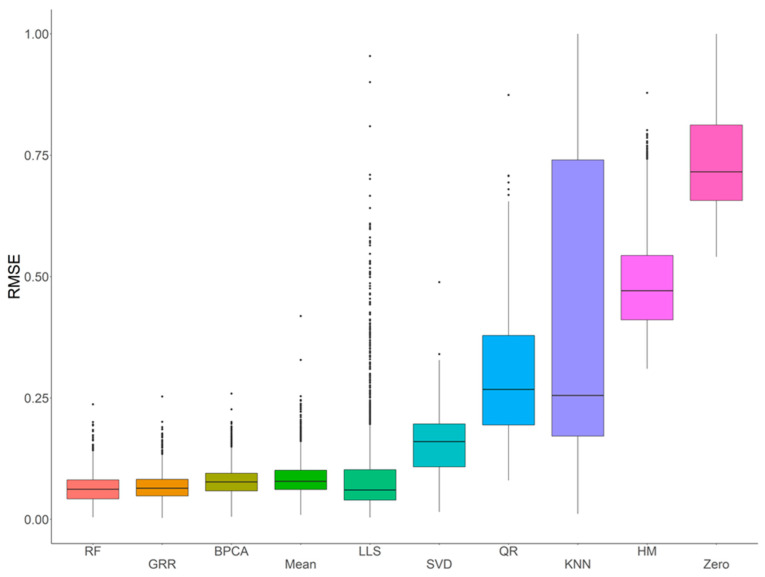
Initial evaluations of imputation accuracy in the complete NIST plasma for a mixture of missingness types (MCAR–MAR–MNAR). Methods are listed across the x-axis, and RMSE is shown on the y-axis. The center line represents the median. The lower and upper box limits represent the 25% and 75% quantiles, respectively. The whiskers extend to the largest observation within the box limit ± 1.5 × interquartile range. Black dots represent single iterations of evaluating RMSE that are outliers. The number of observations for each method is 22,070.

**Figure 3 metabolites-12-00429-f003:**
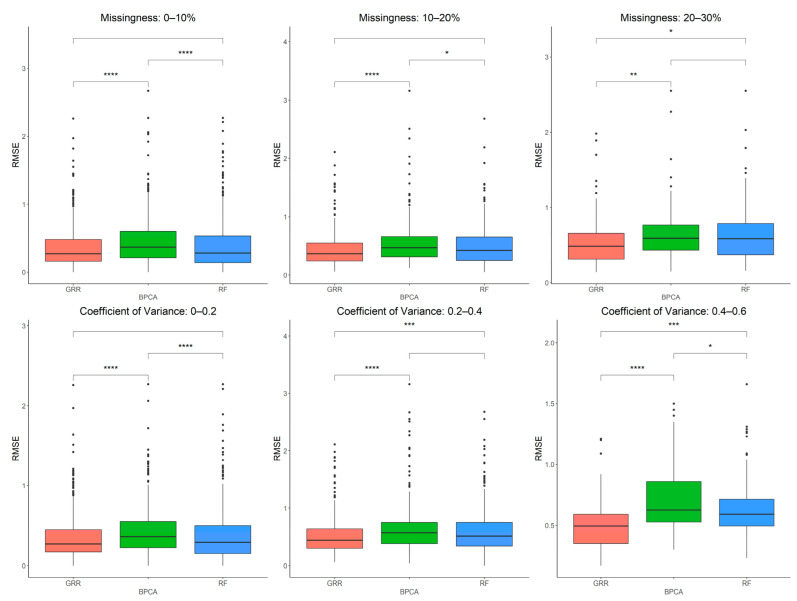
Evaluations of imputation accuracy in the complete baboon plasma. Accuracy is evaluated for levels of missingness types (MCAR–MAR–MNAR). Methods are listed across the x-axis, and RMSE is shown on the y-axis. The top row compares accuracy across a range of missingness types. The bottom row compares accuracy across a range of coefficients of variance. The center line represents the median. The lower and upper box limits represent the 25% and 75% quantiles, respectively. The whiskers extend to the largest observation within the box limit ± 1.5 × interquartile range. Black dots represent single iterations of evaluating RMSE that are outliers. The *p*-values are based on pairwise testing with the Wilcoxon Rank Sum test (* corresponds to *p* ≤ 0.05; ** corresponds to *p* ≤ 0.01; *** corresponds to *p* ≤ 0.001; **** corresponds to *p* ≤ 0.0001).

**Figure 4 metabolites-12-00429-f004:**
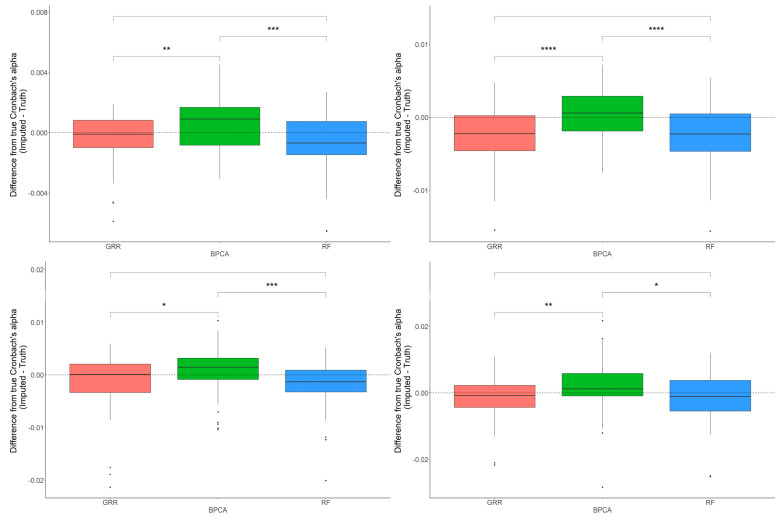
Evaluations of Cronbach’s alpha. Methods are listed across the x-axis, and the difference between the Cronbach’s alphas computed on the complete data and the imputed data is shown on the y-axis. The top row demonstrates differences in Cronbach’s alpha evaluated in the baboon plasma samples for 10% and 20% of overall missingness. The bottom row shows the differences in Cronbach’s alpha evaluated in the baboon liver samples for 10% and 20% overall missingness. The center line represents the median. The lower and upper box limits represent the 25% and 75% quantiles, respectively. The whiskers extend to the largest observation within the box limit ± 1.5 × interquartile range. Black dots represent single iterations of evaluating Cronbach’s alpha that are outliers. The *p*-values are based on pairwise testing with the Wilcoxon Rank Sum test (* corresponds to *p* ≤ 0.05; ** corresponds to *p* ≤ 0.01; *** corresponds to *p* ≤ 0.001; **** corresponds to *p* ≤ 0.0001).

**Figure 5 metabolites-12-00429-f005:**
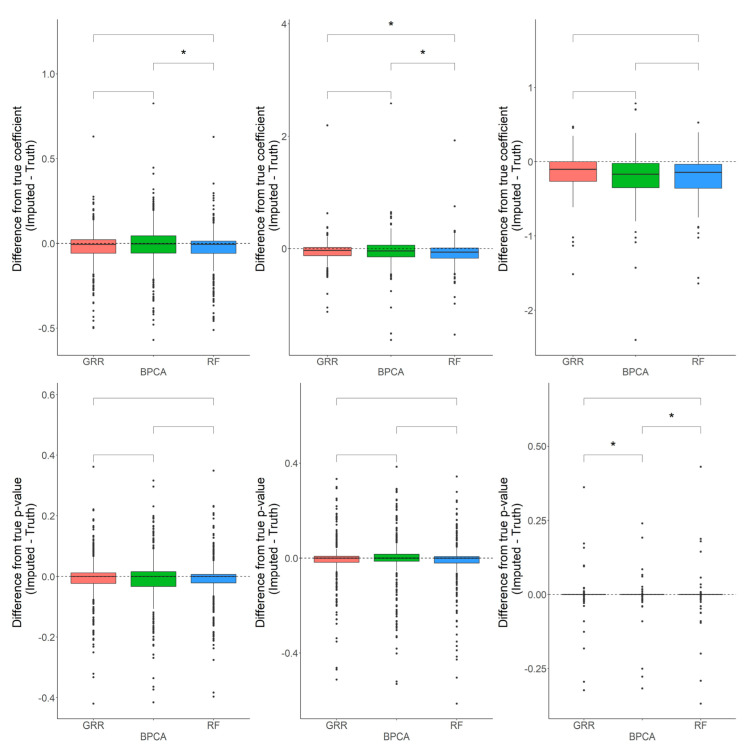
Evaluations of regression coefficient and regression *p*-value accuracy. Methods are listed across the x-axis. The differences between the regression coefficients (or *p*-values) computed on the complete data and the imputed data are shown on the y-axis. The top row demonstrates differences in regression coefficients evaluated in the baboon plasma samples for metabolites with <10%, 10–20%, and 20–30% missingness. The top row demonstrates differences in regression *p*-values evaluated in the baboon plasma samples for metabolites with <10%, 10–20%, and 20–30% missingness. The center line represents the median. The lower and upper box limits represent the 25% and 75% quantiles, respectively. The whiskers extend to the largest observation within the box limit ± 1.5 × interquartile range. Black dots represent single iterations of evaluating differences that are outliers. The *p*-values are based on pairwise testing with the Wilcoxon Rank Sum test (* corresponds to *p* ≤ 0.05).

**Figure 6 metabolites-12-00429-f006:**
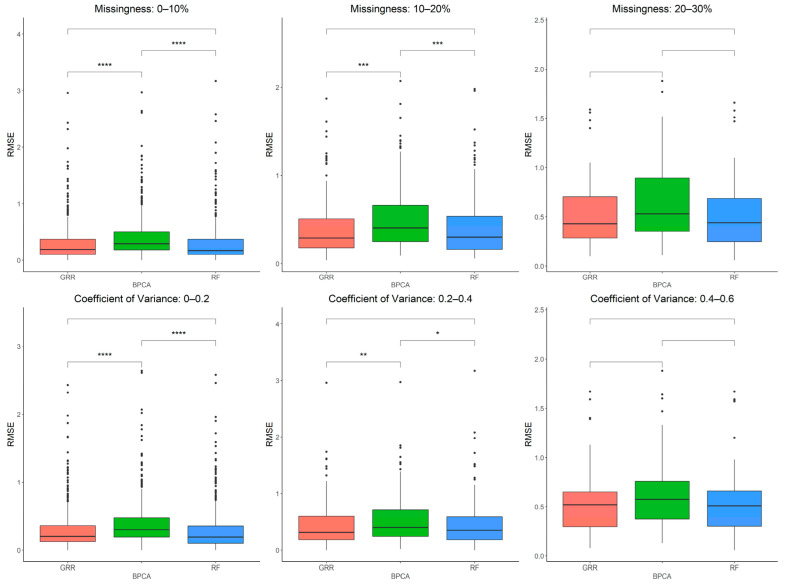
Evaluations of imputation accuracy in the complete baboon liver. Accuracy is evaluated for a mixture of missingness types (MCAR–MAR–MNAR). Methods are listed across the x-axis, and RMSE is shown on the y-axis. The top row compares accuracy across a range of missingness types. The bottom row compares accuracy across a range of coefficients of variance. The center line represents the median. The lower and upper box limits represent the 25% and 75% quantiles, respectively. The whiskers extend to the largest observation within the box limit ± 1.5 × interquartile range. Black dots represent single iterations of evaluating RMSE that are outliers. The p-values are based on pairwise testing with the Wilcoxon Rank Sum test (* corresponds to *p* ≤ 0.05; ** corresponds to *p* ≤ 0.01; *** corresponds to *p* ≤ 0.001; **** corresponds to *p* ≤ 0.0001).

**Figure 7 metabolites-12-00429-f007:**
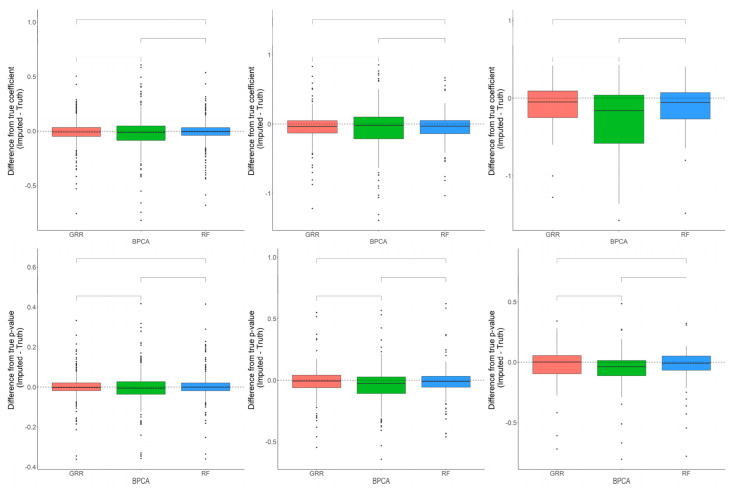
Evaluations of regression coefficient and regression *p*-value accuracy. Methods are listed across the x-axis. The differences between the regression coefficients (or *p*-values) computed on the complete data and the imputed data are shown on the y-axis. The top row demonstrates differences in regression coefficients evaluated in the baboon liver samples for metabolites with <10%, 10–20%, and 20–30% missingness. The top row demonstrates differences in regression *p*-values evaluated in the baboon liver samples for metabolites with <10%, 10–20%, and 20–30% missingness. The center line represents the median. The lower and upper box limits represent the 25% and 75% quantiles, respectively. The whiskers extend to the largest observation within the box limit ± 1.5 × interquartile range. Black dots represent single iterations of evaluating differences that are outliers. The *p*-values are based on pairwise testing with the Wilcoxon Rank Sum test.

**Figure 8 metabolites-12-00429-f008:**
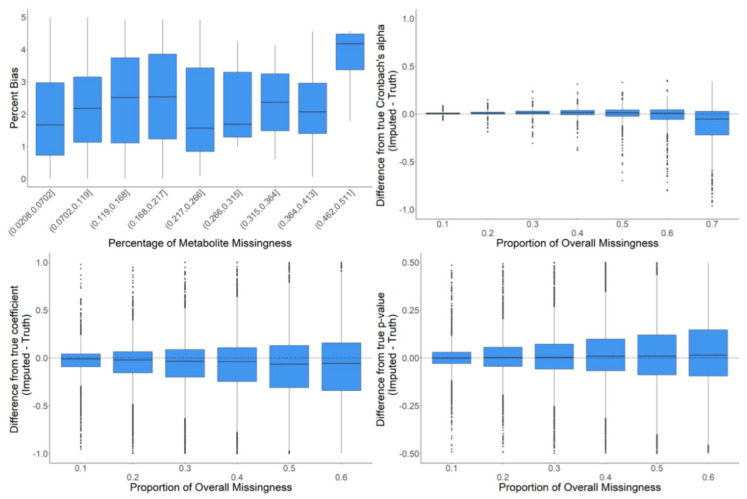
In-depth evaluation of RF imputation. Percent bias (accuracy of imputation on raw data) is shown in the upper left for metabolites in bins of percent missingness between 2–51%. Differences in Cronbach’s alpha are shown in the upper left for a variety of proportions of overall missingness between 10–70%. The differences in regression coefficients are shown in the bottom left for a variety of proportions of overall missingness between 10–60%. The respective differences in *p*-values are shown in the bottom right. The center line represents the median. The lower and upper box limits represent the 25% and 75% quantiles, respectively. Black dots represent single iterations of evaluating differences that are outliers. The whiskers extend to the largest observation within the box limit ± 1.5 × interquartile range.

## Data Availability

The raw data presented in this study are deposited at the metabolomics workbench and available at this link ftp://drccupload@www.metabolomicsworkbench.org/2631/DataTrackID3154 (accessed on 1 April 2022). All spectral libraries are available on request from the corresponding author due to privacy restrictions.

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
