# Peer review of "Optimization of Imputation Strategies for High-Resolution Gas Chromatography–Mass Spectrometry (HR GC–MS) Metabolomics Data"

_metabolites, 2022, doi:10.3390/metabo12050429_

Round 1
Reviewer 1 Report
The review of this submission is hampered by the fact that supplementary figures were not uploaded or were somehow erased or lost. I cannot find them at the submission platform.
Although this could be a minor point, it becomes very important considering the fact that the selection of the three most accurate missing-value imputation methods depends on the simulation of a particular mixture of the 3 main types of missing values with a particular set of proportions and missing-value (global) abundance. This is a study conducted in sequence, and the subsequent analysis depends on the choices made from results presented in figure 1. The authors state (lines 95-96) that “The RMSEs of these methods showed similar patterns across all 4 missing value types (Figures S1-S4).” Although, in the absence of supplementary figures I have no doubt to believe the authors that RF, GRR, BPCA, and Mean show similar patterns for across missing value types, it is essential to have a look at the figures S1 -S4, because, in a scenario of almost 100% MNAR values, previous studies have shown that other LoD related methods outperform at least one of the best of these 4 methods (RF) [ref 11 of the submission]. This case should be taken into consideration, and the downstream analysis performed by the authors might need to include these other methods (half-min, for instance). The results of figure 1 are critical, so it is important to cover all the angles concerning its interpretation and to present figures S1-S4 to support the author's point of view. In my opinion, figures S1-S4, if they show all the methods, deserve to be in the main text of the article.
This paper is well written and easy to follow, and the results support the conclusions put forward in the discussion. However, there is a lot of intersection of this study with that of ref 11 in the manuscript: the general concept and motivation are the same and the authors state that they followed the same simulation strategy for the generation of missing values. Furthermore, when the two studies are compared, as indicated before, the systematic exploration of several scenarios for mixing of each type of missing values seems superficial in this study when compared to ref 11. It is true that the authors diverged from ref 11 by moving forward and exploring replicate correlation as a metric to assess the performance of methods, but this novel approach is hardly a significant contribution.
In line with the lack of significance in the novelty of this submission, it would be important to highlight why this method comparison is particularly tailored to GC-MS (even if HR). In the tile, introduction and from the dataset choice it seems that GC-MS is the particular technique which this comparison applies to, but there is no discussion on why this type of study was necessary for GC-MS since several recent studies on missing-value imputation methods seem to apply in general to most (untargeted) MS techniques. What is so unique to the nature of missing values in GC-MS? It would be important to rigorously clarify this point, as this should be the main motivation behind this work.
As still a major point, this is not really a study about optimization of imputation strategies but the indication of the best method to choose from, and it is hard to find a convenient rule of choice for a given real-life dataset. Related to this concern is the fact that the authors do not mention the optimization of method hyperparameters: some of the methods compared have parameters that can be tuned and optimized. If default values were employed, then the authors should describe software, libraries, and versions for the readers to have an idea how the methods were actually implemented in this study.
Some minor points:
- In legend of fig 2, “The top row compares accuracy across a range of missingness types.”: it is not across types, but rather levels of missingness.
- In figures 2 and 3, giving the high number of points, it is confusing the “swarm” plot on top of the box plots: the box plots would be enough to the understanding of the plot, and it would be make the images cleaner. I suggest something like figures 1 and 4 for figures 2 and 3.
Reviewer 2 Report
The work is interesting and enjoyed reading it. I would recommend this work to be published in Metabolites in the current version however before some very minor corrections as below:
Authors are encouraged to use more examples of employing mass spectrometry-based metabolomics applications in the introduction such as food metabolomics (https://doi.org/10.1016/j.tifs.2008.03.003), forensic medicine (https://www.mdpi.com/2218-1989/11/12/801), environmental metabolomics (https://doi.org/10.1016/j.envint.2021.106503), and precision medicine (https://doi.org/10.1016/j.ejps.2017.05.018).
Line 421: MS-DIAL is not from Agilent Technologies. It is a publicly available software from Riken, Japan, and Fiehn Lab, UC Davis, USA. Please correct it.
Reviewer 3 Report
The authors compared several imputation methods to address the problem of missing values in metabolomics. Although similar comparison was previously performed for LC-MS data (Kokla et al., 2019, 20:492), this study could be also informative because GC-MS is widely used in various metabolomics studies. The analysis is carefully done and the scope of the claim is honestly stated without overstating the case. To make this manuscript clearer and more valuable, I have following suggestions.
- To clarify the significance of this study, authors should state in the introduction why imputation methods need to be compared with GC-MS data. I think it would be good to explain the difference in data characteristics between GC-MS and LC-MS.
- Authors compared 10 imputation methods, but the features are not explained in introduction. How about adding a brief explanation of each method, and their general advantages and disadvantages?
- Section 4.4 describes how to generate missing values, but I could not follow the detailed procedure. Could you please add a detailed and clear explanation? Also, since the systematic evaluation performed in this study is an important aspect, perhaps a brief description of the experimental design could be added before the explanation of results.
Reviewer 4 Report
In this study, the author evaluated the performance of ten commonly used missing value imputation methods with metabolites. By introducing missing values into the complete NIST plasma data set, the author claimed that Random Forest (RF), Glmnet Ridge Regression (GRR), and Bayesian Principal Component Analysis (BPCA) have the lowest root mean square error (RMSE) in the technical replication data. The three methods are further verified in the data of baboon plasma and liver samples, and they all maintain high accuracy. In this research, the author simply compares different methods and selects the best ones, but the author does not put forward his own methods. Therefore, I think the article is not innovative and is not suitable for publication in this journal.
Round 2
Reviewer 1 Report
Considering the revised version of submission “Optimization of Imputation Strategies for High-Resolution Gas 2 Chromatography-Mass Spectrometry (HR GC-MS) Metabolomics Data”, it is clear that the authors have greatly improved the manuscript and have addressed the most serious points raised by the reviewers. Of particular importance:
-The authors have justified why their method comparison approach is suitable to the nature of GC-MS data, developing on previous studies which seemed to be tailored to LC-MS data, which has presumably intrinsically less variability, a statement not universally generalizable, as the nature of samples and the specificity of analytical are also factors that must be considered.
- The authors have provided details on both the missing value simulation procedures and the (hyper)parameters of some of the imputation methods. This improvement in completeness of the methodological sections was mostly welcome.
- The authors have made changes in the figures and figure legends and enhanced the introduction, making the paper easier to follow.
I find no other serious issues, and, in my opinion, this submission does not require a major revision.
However, I would like to suggest that the authors would make some minor changes which, if included, would improve the manuscript. The following two points should be addressed:
- For the pure MNAR simulated data, the “LoD related methods”, such as HM and zero, do not perform well, after all. This contradicts previous results from other authors (for example, refs 11 and 12). I encourage the authors to remark such a fact in the discussion and, possibly, to provide a reason. I do not think that it has to do with the nature of the GC-MS vs LC-MS data, but rather with the consistency of the technical replicates of the NIST plasma data. The nature of the data makes all the other “similarity related” methods perform better, regardless of the simulation procedure adopted to generate MNAR values and the level of missingness.
- RMSE, correlation and Cronbach’s alpha, all behave as measures of internal consistency either by looking at replication in the data or just by simply comparing imputed data with real data but with data with missing values generated from (the real) data with no missing values. Although there is a great appeal from comparing methods using internal consistency measures, a territory much more difficult to chart is the effect of imputation methods on the statistical/machine learning procedures that are employed after the pre-processing stages, such as PCA and PLS-DA. At the end of the day, it is the effect of the imputation method choice on the downstream data analysis stages that matters most from a practical and applied point of view. I encourage the authors to briefly mention and cite some of the studies that go in this direction. The study of ref 12 partially addresses the question, but doi:10.3390/metabo4020433 and doi: 10.3390/metabo11110788 are also worth mentioning.
Reviewer 4 Report
Why not compare the results of the combination of the next two methods, for example, the results of the combination of neural network method and these methods.
